# Surgical resection is sufficient for incidentally discovered solitary pulmonary nodule caused by nontuberculous mycobacteria in asymptomatic patients

Hung-Ling Huang[1,2,3], Chia-Jung Liu[4,5], Meng-Rui Lee[4,5,6], Meng-Hsuan Cheng[2,3], Po-Liang Lu[2,3,8], Jann-Yuan Wang[5,6☯*], Inn-Wen Chong[2,3,7☯*]

1 Kaohsiung Municipal Ta-Tung Hospital, Kaohsiung Medical University Hospital, Kaohsiung, Taiwan,
2 Department of Internal Medicine, Kaohsiung Medical University Hospital, Kaohsiung, Taiwan, 3 Graduate Institute of Medicine, College of Medicine, Kaohsiung Medical University, Kaohsiung, Taiwan, 4 Department of Internal Medicine, National Taiwan University Hospital, Hsin-Chu Branch, Hsin-Chu, Taiwan, 5 Department of Internal Medicine, National Taiwan University Hospital, Taipei, Taiwan, 6 National Taiwan University, College of Medicine, Taipei, Taiwan, 7 Departments of Respiratory Therapy, Kaohsiung Medical University Hospital, Kaohsiung, Taiwan, 8 Department of Laboratory Medicine, Kaohsiung Medical University Hospital, Kaohsiung, Taiwan

☯ These authors contributed equally to this work.
* chong@kmu.edu.tw (IWC); jywang@ntu.edu.tw (JYW)

## Abstract

Incidentally discovered solitary pulmonary nodules (SPN) caused by nontuberculous mycobacteria (NTM) is uncommon, and its optimal treatment strategy remains uncertain. This cohort study determined the clinical characteristics and outcome of asymptomatic patients with NTM-SPN after surgical resection. Resected SPNs with culture-positive for NTM in six hospitals in Taiwan during January, 2010 to January, 2017 were identified. Asymptomatic patients without a history of NTM-pulmonary disease (PD) or same NTM species isolated from the respiratory samples were selected. All were followed until May 1, 2019. A total of 43 patients with NTM-SPN were enrolled. *Mycobacterium avium* complex (60%) and *M. kansasii* (19%) were the most common species. The mean age was 61.7 ± 13.4. Of them, 60% were female and 4% had history of pulmonary tuberculosis. The NTM-SPN was removed by wedge resection in 38 (88%), lobectomy in 3 (7%) and segmentectomy in 2 (5%). Caseating granuloma was the most common histologic feature (58%), while chronic inflammation accounts for 23%. Mean duration of the follow-up was 5.2 ± 2.8 years (median: 4.2 years [2.5–7.0]), there were no mycobacteriology recurrence or NTM-PD development. In conclusion, surgical resection is likely to curative for incidentally discovered NTM-SPN in asymptomatic patients without culture evidence of the same NTM species from respiratory specimens, and routine mycobacterium culture for resected SPN might be necessary for differentiating pulmonary tuberculosis and NTM because further treatment differs.

**Data Availability Statement:** All relevant data are within the paper.

**Funding:** This study was supported by the Taiwan Ministry of Science and Technology (MOST107-2314-B-037-106-MY3), Ministry of health and welfare (MOHW108-TDU-B-212-133006) and Kaohsiung Medical University Hospital Research Program (KMUH107-7R12). The funders had no role in study design, data collection and analysis, decision to publish, or preparation of the manuscript.

**Competing interests:** The authors have declared that no competing interests exist.

# Introduction

With advances in and the widespread use of chest computerised tomography (CT), the detection rate of solitary pulmonary nodules (SPNs) has increased in recent decades, and the detection rate exceeded 50% among asymptomatic patients in some series [1, 2]. The etiology of SPN varies substantially among various studies. A positive correlation between nodule size and the likelihood of malignancy has been demonstrated. According to the Fleischner Society guideline and Mayo Clinic CT Screening Trial, the likelihood of malignancy is less than 1% for nodules less than 6 mm in diameter but up to 50% for those larger than 20 mm [3, 4]. A population-based study conducted in the United States and United Kingdom revealed that the vast majority of SPNs are benign, even in smokers [5].

Granuloma due to infection is the most common aetiology of benign SPN [6], with the exact proportion depending on the geographic location. In a study including 201 SPNs in South Korea, an endemic area for tuberculosis (TB), tuberculoma accounted for 60% of benign SPNs, but none of them yielded nontuberculous mycobacteria (NTM) [7]. In a study in Japan, among 25 benign lesions in 103 surgically removed SPNs, the responsible microorganism was *Mycobacterium tuberculosis* in 1 (4%) case and *Mycobacterium avium* complex (MAC) in 4 (16%) cases [8]. In a study conducted in the United States, only 1 (5%) of the 20 resected SPNs was culture-positive for *M. tuberculosis*, whereas 14 (70%) were positive for NTM [9]. In NTM-SPNs, MAC is the most commonly isolated species (70%–89%) [2, 10–14].

Limited data are available on the treatment of NTM-SPN. The current guidelines suggest that surgical resection without postoperative antibiotics may be sufficient for treating SPN caused by MAC if there is no other radiographic abnormality (poor quality of evidence, Grade D) [15, 16]. The only reported evidence is from a retrospective study conducted in 126 patients with chronic obstructive pulmonary disease (COPD) who underwent lung volume reduction surgery. Among the 142 lung specimens, 14 exhibited caseating granuloma, implying mycobacterial infection. Only three specimens were culture-positive for NTM. In addition, no follow-up imaging and mycobacteriology studies were performed [17].

Because of the globally increasing prevalence of NTM disease [16, 18], the increasing use of low-dose chest CT for routine health examination in Taiwan [19], and the high availability and safety of video-assisted thoracic surgery for SPN resection [20], physicians will encounter incidentally discovered NTM-SPNs more frequently in the future. Reliable evidence confirming the most appropriate treatment strategy for NTM-SPN is required. We therefore conducted this multicentre, retrospective, longitudinal follow-up study of asymptomatic patients with incidentally discovered NTM-SPN to determine their clinical characteristics and outcomes after surgical resection.

# Material and methods

## Study population

This retrospective, longitudinal study was conducted at two medical centres—the National Taiwan University Hospital (NTUH) and Kaohsiung Medical University Hospital (KMUH)—and their four branch hospitals between January 2010 and January 2017. The study was approved by the institutional review boards (IRB) of the two centres (NTUH REC 201508017RIND and KMUH IRB-SV[I]-2015200266), and IRB waived the need for informed consent because data utilized in this retrospective study have been de-identified.

Lung tissues obtained after surgical resection were retrieved from mycobacteriology databases. Patients whose resected tissues were culture-positive for NTM were identified. Among them, those with multiple pulmonary nodules or consolidation confirmed by chest CT were

excluded and only patients with SPNs were selected. For all respiratory specimens, acid-fast smear (AFS), mycobacterial culture, and species identification were performed using previously reported methods [18]. SPN was defined as a small (≤3 cm), well-defined lesion completely surrounded by aerate lung parenchyma on chest CT [3].

Patients were further excluded if (1) they were symptomatic and their NTM-SPNs were not incidentally discovered; (2) PD caused by NTM had been diagnosed previously or was diagnosed at present according to the 2007 guidelines of the American Thoracic Society and Infectious Disease Society of America [15]; (3) same sputum sample was culture-positive for the NTM species isolated from the SPN before resection; (4) they had active pulmonary TB or malignancy concomitantly before pulmonary resection; (5) demographic data were missing; and (6) the duration of follow-up was less than 24 months after resection.

### Data collection

For each patient, the following data were collected: age, sex, body mass index (BMI), smoking status, underlying diseases, initial manifestations, SPN size and location on chest CT, preoperative lung function, laboratory data, surgical methods (segmentectomy, wedge resection, or lobectomy), mycobacteriological and histopathological findings of the lung specimen, results of follow-up sputum, latest chest image (CT preferred) before the end of follow-up, treatment course, and survival. Sputum AFS and mycobacterial culture were considered negative if patients failed to expectorate any sputum even after sputum induction [21]. The effective standard anti-NTM treatment was based on current NTM guidelines [15, 16]. Chest images were interpreted by two pulmonologists blinded to the clinical data. Discrepancies were resolved through consensus. All patients were followed up until May 1, 2019 or death. Duration of follow-up was calculated by three different ways: 1) operation to last date of chest CT follow-up; 2) operation to last date of chest x-ray follow-up; and 3) operation to last date of hospital visit.

### Outcomes

The primary outcome was the incidence of NTM-PD during the follow-up after resection of SPN. The secondary outcome was the incidence of new lesion on follow-up chest images and bacteriological recurrence in follow-up sputum mycobacterial studies.

### Statistical analysis

All statistical analyses were performed using SPSS (version 22.0; SPSS Inc., Chicago, IL, USA). Intergroup differences were determined using the independent sample $t$ test for continuous variables and the *chi*-square test or Fisher's exact test for categorical variables, as appropriate. Two-side $p$ values of $<0.05$ were considered statistically significant.

## Results

### Selection of study participants

Fig 1 shows the flowchart of patient selection and the enrolment criteria. Between 2010 and 2017, 248 lung specimens yielding NTM from 214 cases were retrieved from the mycobacteriology databases of six hospitals. After applying a serial of exclusion criteria mentioned above, a total of 43 resected NTM-SPNs from 43 cases were finally selected for analysis. The follow-up duration of each patient was at least two years since resection, with the mean of 5.2 ± 2.8 years.

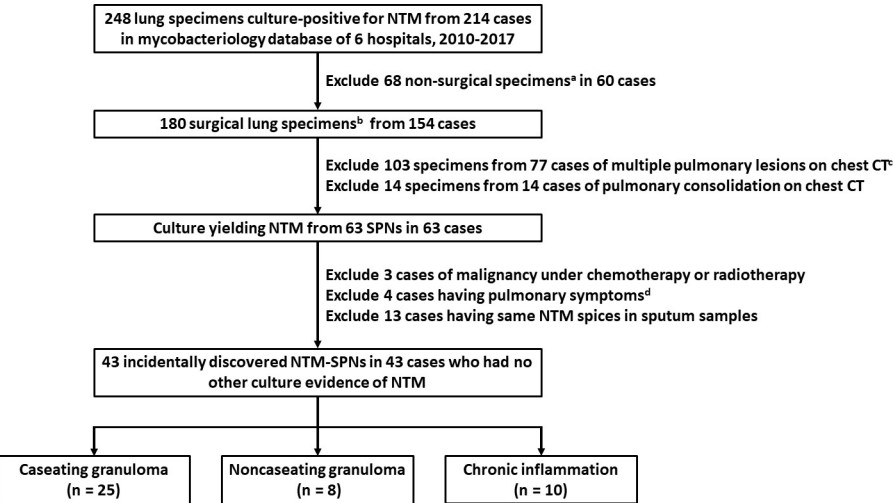

Fig 1. Case selection process. (CT, computerised tomography; NTM, nontuberculous mycobacteria; PD, pulmonary disease; SPN, solitary pulmonary nodule) [a] Including echo-guided biopsy, bronchoscopic biopsy, and computerised tomography-guided biopsy. [b] Including 131 specimens obtained by wedge resection, 40 by lobectomy, and 9 by segmentectomy. [c] Including 66 specimens from 48 cases of multiple lung nodules, 14 specimens from 10 cases of multiple consolidations, and 23 specimens from 18 cases of other radiographic patterns, such as nodular infiltrations, bronchiectasis, ground-glass opacities, and atelectasis. [d] The symptom was productive cough in four and hemoptysis in the other three.

## Clinical characteristics of patients

The clinical characteristics of the 43 patients with NTM-SPN are summarised in Table 1. The mean age was 61.7 ± 13.4 years, with a female/male ratio of 1.5. Among them, 88% had never smoked and 3 (7%) were malnourished (BMI < 18.5 kg/m$^2$). The most common underlying diseases were the complete remission of malignancy after treatment (n = 11, 26%), including 5 patients with early stage lung cancer, 3 with colorectal cancer, 2 with breast cancer and 1 with prostate cancer. Six (14%) patients had the underlying disease of diabetes mellitus (DM, 14%). None received serology test for *Human Immunodeficiency Virus* (HIV) infection.

All patients were asymptomatic, and their SPNs were discovered incidentally during a health examination or routine work-up for underlying comorbidities. Five patients (10%) had impaired lung function. The laboratory data were within normal ranges.

## Radiographic features and characteristics of SPN

Table 2 summarises the characteristics of NTM-SPNs. The most common locations of resected SPNs were the right upper lobe (32%) and left upper lobe (16%). The mean diameter of SPNs was 1.8 ± 1.0 cm, nine (21%) of the SPNs were cavitary lesions and two of them had the calcification (5%). Positron emission tomography (PET) was performed in 16 patients, with the maximal standardised uptake value (SUVmax) being 6.4 ± 7.2 (median: 4.2; IQR: 2.0–5.9).

The surgical method of SPN removal was wedge resection in 38 (88%), lobectomy in 3 (7%), and segmentectomy in the remaining 2 patients (5%). The result of lung tissue AFS was positive in 9 (21%) patients. The most common mycobacterial isolate of the 43 SPNs was MAC (n = 26, 60%), followed by *M. kansasii* (8, 19%). No surgical complications occurred. Among patients with SPN, 33 (77%) had histological findings suggestive of mycobacterial infection, including 25 (58%) and 8 (19%) showed caseating granulomatous inflammation and noncaseating granulomatous inflammation, respectively.

**Table 1. Clinical characteristics of the 43 patients with nontuberculous mycobacteria (NTM)-solitary pulmonary nodule.**

| Variables | N = 43 |
|---|---|
| Age (year) | 61.7 ± 13.4 |
| Female sex | 26 (60%) |
| Body-mass index (kg/m$^2$) | 22.6 ± 3.5 |
| <18.5 | 3 (7%) |
| Smoking status | |
| Current | 3 (7%) |
| Former | 2 (5%) |
| Underlying diseases | |
| History of pulmonary tuberculosis | 2 (5%) |
| Bronchiectasis | 1 (2%) |
| Chronic obstructive pulmonary disease | 1 (2%) |
| Asthma | 2 (5%)[a] |
| Lung cancer under complete remission | 6 (14%)[b] |
| Other cancer under complete remission | 5 (12%)[c] |
| Diabetes mellitus | 6 (14%) |
| Rheumatoid arthritis | 2 (4%)[d] |
| Baseline laboratory data | |
| Leukocyte (K/uL) | 6.1 ± 1.7 |
| Hemoglobin (g/dL) | 13.0 ± 1.6 |
| Platelet count (K/uL) | 222.8 ± 55.1 |
| Aspartate transaminase (U/L) | 23.2 ± 6.9 |
| Alanine transaminase (U/L) | 19.4 ± 9.9 |
| Total bilirubin (mg/dL) | 0.6 ± 0.2 |
| Creatinine (mg/dL) | 0.9 ± 0.3 |
| Albumin (g/dL) (n = 48) | 4.3 ± 0.4 |
| CEA (n = 42) | 1.9 ± 1.7 |
| Baseline lung function (n = 46) | |
| FEV$_1$ (% of predicted) | 110.5 ± 22.1 |
| FEV$_1$/FVC | 81.3 ± 4.9 |
| Obstructive defect | 1 (2%) |
| Restrictive defect | 4 (9%) |

Abbreviation: CEA, carcinoembryonic antigen; FEV$_1$, forced expiratory volume in one second; FVC, forced vital capacity.

Data are either number (%) or mean ± standard deviation.

[a] One patient had severe asthma under long-term steroid therapy. The other patient received prednisolone at 15 mg per day for 6 months and used long-term inhaled corticosteroids.

[b] Histological reports of resected lung tissue demonstrated lung adenocarcinoma in five patients and another one is squamous cell carcinoma.

[c] Three patient had colorectal cancer, one patient had breast cancer, and one patient had prostate cancer.

[d] Both patients received long-term disease-modifying antirheumatic drugs and steroids.

## Sputum mycobacteriology study

Mycobacteriological study of either sputum or bronchoalveolar lavage was performed before surgery in 36 (84%) of the 43 patients with NTM-SPN (Table 3). Of these samples, the culture results from bronchoscopic specimens were positive for MAC in one case with *M. kansasii*-SPN and *M. gordonae* in another with MAC-SPN.

**Table 2. Operation methods, results of tissue histopathology, and mycobacteriology studies of the 43 patients with nontuberculous mycobacteria (NTM) solitary pulmonary nodule.**

| Variables | N = 43 |
|---|---|
| Preoperative radiographic features | |
| Location: right side | 29 (67%) |
| Right upper lobe | 14 (32%) |
| Right middle lobe | 7 (16%) |
| Right lower lobe | 8 (19%) |
| Location: left side | 14 (33%) |
| Left upper lobe | 7 (16%) |
| Left lower lobe | 7 (14%) |
| Cavitation | 9 (21%) |
| Calcification | 2 (5%) |
| Size (cm) | 1.8 ± 1.0 |
| PET-CT, SUVmax (n = 16) | 6.4 ± 7.2 |
| Operation method | |
| Wedge resection | 38 (88%) |
| Lobectomy | 3 (7%) |
| Segmentectomy | 2 (5%) |
| Histopathology | |
| Caseating granulomatous inflammation | 25 (58%) |
| Non-caseating granulomatous inflammation | 8 (19%) |
| Chronic inflammation | 10 (23%) |
| Tissue mycobacteriology study | |
| No. of AFS-positive case | |
| Grade 1 or 2 | 7 (16%) |
| Grade 3 or 4 | 2 (5%) |
| Culture | |
| MAC | 26 (60%) |
| *M. kansasii* | 8 (19%) |
| *M. abscessus* | 3 (7%) |
| *M. fortuitum* | 2 (5%) |
| *M. chelonae* | 1 (2%) |
| Unidentified NTM | 3 (7%)[a] |

Abbreviations: AFS, acid-fast smear; MAC, *Mycobacterium avium* complex; PET-CT, positron emission tomography–computed tomography; SUVmax, maximal standardised uptake value.

Data are represented as either number (%) or mean ± standard deviation.

[a] The histological reports of the 3 resected SPNs all indicated granulomatous inflammation. Tissue mycobacterial culture yielded photochromogens in 2 and nonphotochromogens in the remaining one. However, species identification was not possible under routine laboratory procedures.

During follow-up, 24 (56%) patients had no or scanty sputum even upon sputum induction and 6 of them only ever received bronchoscopic samplings thereafter. None of the follow-up respiratory samples yielded the same NTM species as the initial tissue culture. Two isolates of *M. gordonae* were identified from bronchoscpic specimens in two patients.

## Outcome status

A total of four, three with MAC-SPN and one with *M. kansasii*-SPN ever received standard anti-TB treatment after surgery, with a treatment duration of 160 ± 44.6 days (Table 4). None

**Table 3. Preoperative and follow-up sputum mycobacteriology study and preoperative radiological features of the 43 patients with nontuberculous mycobacteria (NTM)-solitary pulmonary nodule (SPN).**

| Variables | N = 43 |
|---|---|
| Pre-operative mycobacteriology study from respiratory tract | |
| Bronchoscopic specimens (n = 12) | 12 |
| Number of expectorated sputum samples per case (n = 24) | 1.8 ± 1.6 |
| Number of cases with positive sputum AFS | 0 |
| Culture-positive for NTM other than that isolated from SPN | 2 (5)[a] |
| Follow-up mycobacteriology study from respiratory tract (n = 19)[b] | |
| Bronchoscopic specimens | 6 |
| Number of expectorated sputum samples per case | 1.9 ± 0.7 |
| Timing of follow-up (years after operation) | 3.3 (1.2) |
| No. of AFS-positive case | 0 |
| Culture-positive for NTM | 2 (5) |
| *M. gordonae* | 2 |
| Same species as the tissue culture | 0 |

Abbreviation: AFS, acid-fast smear; MAC, *Mycobacterium avium* complex.

Data are number (%), mean ± standard deviation or median (IQR).

[a] The culture results from bronchoscopic specimen were positive for MAC in one case with *M.kansasii*-SPN and *M. gordonae* in another with MAC-SPN.

[b] 24 patients failed to produce any sputum even upon sputum induction using hypertonic saline solution and were considered as sputum culture-negative for NTM.

**Table 4. Treatment outcomes of the 43 patients with nontuberculous mycobacteria (NTM)-solitary pulmonary nodule (SPN).**

| Variables | N = 43 |
|---|---|
| Ever received any anti-mycobacterial drug | 4 (9) [a] |
| Received effective chemotherapy against NTM-PD >6 months | 0 |
| Follow-up duration (years after operation) | |
| To last date of chest CT follow-up (n = 35) | 2.5 ± 1.9 |
| To last date of chest x-ray follow-up (n = 43) | 3.8 ± 2.1 |
| To last date of hospital visit (n = 43) | 5.2 ± 2.8 |
| Outcome | |
| Mortality | 0 |
| New pulmonary lesion | 1 (2) |
| Mycobacteriology recurrence | 0 |
| Fulfilling the diagnosis of NTM-PD | 0 |

Abbreviation: PD, pulmonary disease.

Data are expressed as either number (%) or mean ± standard deviation.

[a] Three cases had *Mycobacterium avium* complex–SPN, and the remaining one had *M. kansasii*-SPN. The primary care physicians decided to complete a full course of antituberculosis treatment (160 ± 44.6 days) based on the histological finding of granulomatous inflammation and positive acid-fast smear of resected SPNs. All the 4 patients were clinically asymptomatic.

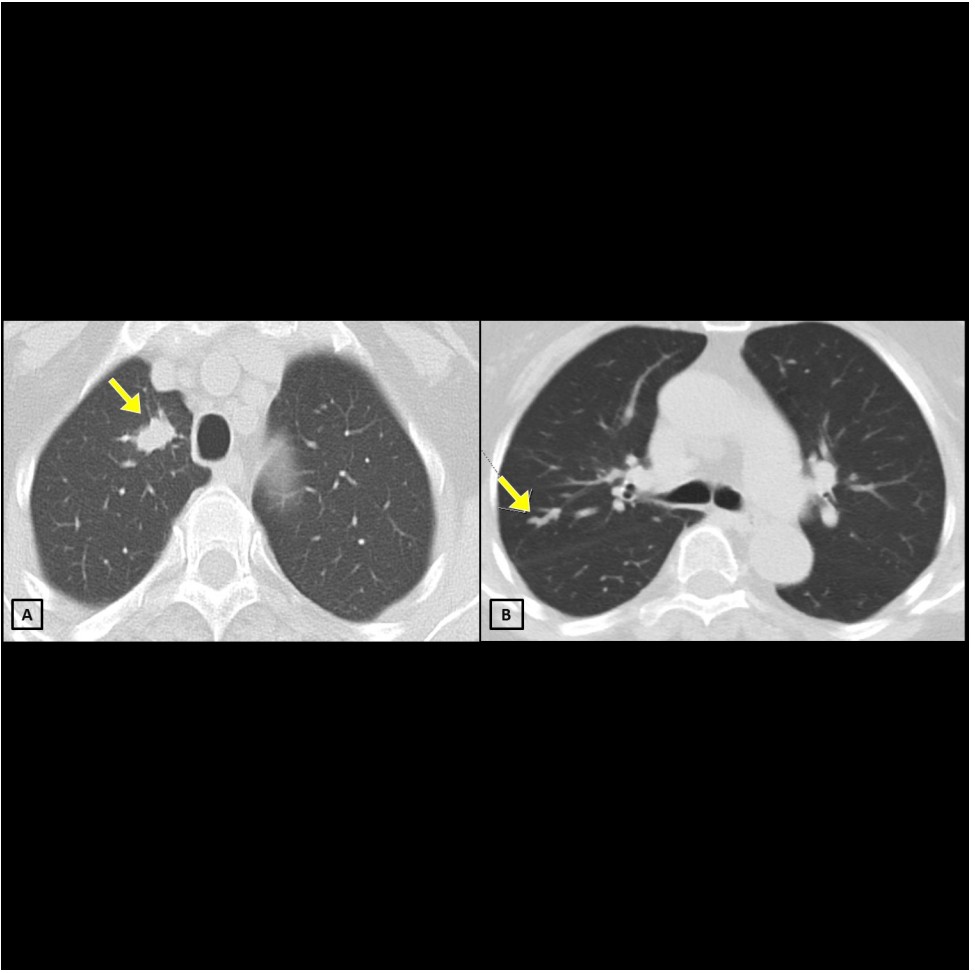

**Fig 2. A) Chest computerised tomography (CT) of a 79-year-old asymptomatic woman before wedge resection** shows a $1.7 \times 1.0$ cm$^2$ lobulated nodule over the right upper lobe (RUL). The tissue culture yielded *Mycobacterium avium* complex. She remained asymptomatic and did not receive medical treatment after resection. **B) CT performed 9 months later shows new nodular infiltrates over the RUL.** In the subsequent 2 years, the patient remained asymptomatic. The subsequent mycobacterial cultures from 3 expectorated sputum samples and 1 bronchoscopic sample were all negative for NTM.

of the patients received effective anti-NTM treatment for more than 6 months. During the mean duration of follow-up $5.2 \pm 2.8$ years (median: 4.2 years [2.5–7.0]), there were no mortalities, and NTM-PD was not diagnosed in any patient because of the lack of suggestive radiographic and microbiologic evidence.

In one patient who underwent wedge resection of the right upper lobe for MAC-SPN, a new lesion with nodular infiltration was noted in the same lobe but different location as previous one on chest CT at 9 months after the operation. She remained asymptomatic during this period (Fig 2). The subsequent mycobacterial cultures from 3 expectorated sputum samples and 1 bronchoscopic sample were all negative for NTM during a 2-year follow-up.

## Discussion

This is the first retrospective study to investigate the long-term outcomes of asymptomatic patients with incidentally discovered NTM-SPN after surgical resection. Two main findings

were obtained. First, surgical resection is curative for NTM-SPN in asymptomatic patients without positive culture of the same NTM species from respiratory specimens and a history of NTM-PD. Further medical treatment for NTM will probably not be necessary. Second, MAC and *M. kansasii* accounted for 79% of the resected SPNs.

The detection rate of SPN is increasing because of the increasing use of chest CT in current clinical practice and the performance of routine health check-ups. In one study, granuloma was discovered to be the most common cause of benign SPNs and was usually, if not always, assumed to be TB related [6]. NTM pulmonary infection has been demonstrated to present as SPN [11, 15, 22], with an incidence of 2% in a study conducted in South Korea [22]. The true incidence and outcome of NTM-SPN remain unclear because bacteriological confirmation of resected lung tissue has been lacking in most studies [9, 17, 22–25], and sometimes both patients with SPN and those with other radiographic patterns of lung lesions with multiple involvement have been enrolled [22, 26]. According to a literature review, we found only 74 cases of culture-confirmed and pure NTM-SPN, which were identified in six studies (Table 5) [9, 17, 22, 25–27]. Although the duration of follow-up in these studies varied from 0 to 10 years, the overall mean duration was less than 2 years. In addition, crucial information was frequently missing. Medical treatment was administered to only 19 (26%) patients, usually a regimen against TB rather than NTM [9, 26]. Nevertheless, in only 2 of the 74 cases (one of them received anti-NTM treatment after resection), recurrent nodular infiltration was discovered at the other lobe, and no bacteriological evidence of NTM recurrence was obtained [22]. Because of the heterogeneity in patient selection criteria and clinical evaluation, small sample size, and relative short follow-up duration of the above-mentioned studies, reliable recommendations for the management of NTM-SPN remain to be determined.

Anti-TB treatment lasting 6–12 months is routinely administered after surgery for pulmonary granuloma caused by *M. tuberculosis* [28]. Because of the global trend of aging, the high risk of toxicity due to "empiric anti-TB treatment" [29] should be carefully balanced with potential benefit of the treatment. In a retrospective study, 67 patients with lung nodule(s) and histopathological findings suggestive of TB but lacking microbiologic confirmation were followed up, and they were not immediately administered anti-TB treatment [30]. Within 3.7 ± 0.7 years, only one, a cancer patient under regular chemotherapy and target therapy, developed active TB, corresponding to an incidence rate of 4 cases/1000 person-years. Although NTM is less virulent than *M. tuberculosis*, its multidrug treatment is even more toxic and lengthy [31, 32]. Therefore, determining the necessity of postoperative anti-NTM treatment for patients with completely resected NTM-SPN is of practical importance. In this study, which can be considered the largest longitudinal study in NTM-SPN, none of the 43 patients with NTM-SPN developed NTM-PD within 5.2 ± 2.8 years, suggesting that conservative management with regular follow-up, rather than immediate chemotherapy, may be sufficient for treating NTM-SPN.

In this study, MAC was the predominant causative organism of NTM-SPN (60%), which is in compatible with previous reports [9, 11, 22, 25, 26, 33]. Interestingly, *M. kansasii* was more common in this study cohort than in those of previous reports [26, 27], accounting for 19% of the SPNs. This might be related to increasing *M. kansasii* isolation in Taiwan [18, 34].

During follow-up, 2 patients had subsequent different NTM isolates from sputum as initial tissue culture. The single NTM isolate from respiratory tract should be interpreted with caution under consideration of the colonization rather than true pathogen for patients with NTM-SPN [15, 26].

This study has some limitations. First, this retrospective study lacked a standardised treatment and follow-up protocol. Second, a resected SPN was sent for mycobacterial culture at the discretion of the surgeon. This might have resulted in underestimation of the incidence of

**Table 5. Literature review of the clinical characteristics and postoperative outcomes of patients with solitary pulmonary nodule (SPN) caused by nontuberculous mycobacteria (NTM).**

| Case No.[Ref] | Underlying disease | Male sex | Age[a] (year) | No symptom | Histology | SPN size (cm) | Main NTM spp. (%) | Same NTM spp. in sputum | Medical Tx[a] (%) | FU duration (year) | Recurrent case (%) |
|---|---|---|---|---|---|---|---|---|---|---|---|
| 14 [9] | n.a. | 4 | 64 (53–70) | 100% | GR | n.a. | MAC (86%) | 1 | 10 (71%)[b] | 1.7 (1.0–10) | 0 |
| 24 [25] | Old TB: 2; Cancer: 2 | 8 | 64 (49–77) | 96% | GR | 0.7–3.0 | MAC (100%) | 0 | 0% | Mean: 0.7 | 0 |
| 2 [17] | COPD | n.a. | n.a. | 0 | CG | n.a. | *M. xenopi* (100%) | 0 | n.a. | n.a. | n.a. |
| 28 [26] | n.a. | 12 | 59 (33–79) | 89% | n.a. | n.a. | MAC (96%) | 2 | 9 (32%) | Mean: 1.8 | 2 (7%)[c] |
| 5 [22] | n.a. | n.a. | n.a. | n.a. | GR | 1.3–3.4 | n.a. | 3 | 0 | 0.5–6 | 0 |
| 1 [27] | 0 | 1 | 46 | 100% | GR | 2.5 | *M. kansasii* | 0 | 0 | 0.5 | 0 |

Abbreviations: CG, caseating granuloma; CI, chronic inflammation; COPD, chronic obstructive pulmonary disease; FU, follow-up; GR, granuloma; n.a., not available; OP, operation; Tx, treatment.

[a] Data are expressed as median (range) unless otherwise mentioned.

[b] The 10 patients received anti-tuberculosis treatment.

[c] Both had recurrent nodule at different lobes from the previous surgical site, but none underwent mycobacterial culture for the recurrent nodule. One received medical treatment for NTM after operation.

NTM-SPN. Third, 4 patients received the anti-TB medication after surgical resection, which may be a potential confounder to alter the subsequent outcome. Lastly, because serology test for HIV infection was not checked and few patients received steroids or immunomodulating agents, the impact of immunosuppression, either disease-associated or iatrogenic, on the risk and clinical course of NTM-SPN remains to be studies.

In conclusion, regular follow-up rather than immediate administration of toxic and lengthy chemotherapy for incidentally discovered NTM-SPN after surgical resection should be the optimal management strategy for asymptomatic patients without culture evidence of the same NTM species from respiratory specimens. Considering the increasing prevalence of NTM-related disease worldwide and the improving detection of SPNs, routine mycobacterial culture for resected SPN could be considered for differentiating TB and NTM because their treatment differs.

## Acknowledgments

The authors thank the Information Technology Office of National Taiwan University Hospital and Statistical Analysis Laboratory, Department of Medical Research, Kaohsiung Medical University Hospital, for providing patient data.

## Author Contributions

**Conceptualization:** Hung-Ling Huang, Jann-Yuan Wang, Inn-Wen Chong.

**Data curation:** Hung-Ling Huang, Chia-Jung Liu, Meng-Hsuan Cheng.

**Formal analysis:** Hung-Ling Huang.

**Funding acquisition:** Hung-Ling Huang.

**Investigation:** Meng-Rui Lee.

**Methodology:** Hung-Ling Huang, Chia-Jung Liu, Meng-Rui Lee.

**Project administration:** Hung-Ling Huang, Chia-Jung Liu, Meng-Rui Lee, Meng-Hsuan Cheng.

**Supervision:** Po-Liang Lu, Jann-Yuan Wang, Inn-Wen Chong.

**Validation:** Po-Liang Lu, Jann-Yuan Wang.

**Writing – original draft:** Hung-Ling Huang.

**Writing – review & editing:** Jann-Yuan Wang, Inn-Wen Chong.

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
