## [Decision Letter · Decision Letter 0]

20 Aug 2019

PONE-D-19-16108

Surgical resection is sufficient for incidentally discovered solitary pulmonary nodule caused by nontuberculous mycobacteria in asymptomatic patients

Dear Dr Inn-Wen Chong

Thank you for submitting your manuscript to PLOS ONE. After careful consideration, we feel that it has merit but does not fully meet PLOS ONE’s publication criteria as it currently stands. Therefore, we invite you to submit a revised version of the manuscript that addresses the points raised during the review process.

To enhance the reproducibility of your results, we recommend that if applicable you deposit your laboratory protocols in protocols.io, where a protocol can be assigned its own identifier (DOI) such that it can be cited independently in the future. For instructions see: http://journals.plos.org/plosone/s/submission-guidelines#loc-laboratory-protocols

We look forward to receiving your revised manuscript.

Kind regards,

Adriano Gianmaria Duse, MD

Academic Editor

PLOS ONE

Journal Requirements:

**Comments to the Author**

Many thanks for the submission of this manuscript. The manuscript is technically sound and well-written. Current practice for patients with asymptomatic solitary pulmonary nodules caused by NTMs is generally to consider that resection is sufficient treatment, but there are limited data to support this practice. While this is a retrospective study, it is unlikely RCTs will be performed given the relative rarity of this condition, and the data do add to current knowledge.

We kindly request you to correct the grammatical errors listed below as well as to address the comments of the reviewers.

Grammatical / typographical errors: 

Introduction 

"tuberculoma accounted for 60% of benign SPNs, but none of them yielding nontuberculous mycobacteria (NTM)". Suggest "...none of them yielded..." 

Methods: 

"and their four branch hospitals between January 2010 and Jnauary 2017." 

Correct spelling 

Results: 

"During follow-up, 24 (56%) patients had no or scanty sputum even upon sputum

induction and 6 of them ever received bronchoscopic samplings thereafter" 

Not clear what "...ever received bronchoscopic samplings" means. Maybe "...6 of them only ever received bronchoscopic sampling thereafter"? 

Discussion: 

Further medical treatment for NTM will probably not necessary. Suggest "..not be necessary". 

This might be resulted in underestimation of the incidence of NTM-SPN. Suggest "...might have resulted in..."

A few minor clarifications are required:

1) The method of case ascertainment was not entirely clear. Based on the figure it seems that the authors examined a database containing all positive cultures for NTM, and then looked for patients with resected solitary lung nodules. Additional detail regarding the exact methods used to identify potential cases would be helpful. 

One of the exclusion criteria was the same sputum sample being culture-positive for the NTM species isolated from the SPN before resection. What do you mean by "same sputum sample" - is this a sputum sample taken at the same time as the surgery? Did you define a time period relative to the surgery when the sputum sample could/should have been taken? 

2) As this was a retrospective cohort study, follow-up methods were not standardized for all patients, which may have led to under-ascertainment of recurrent nodules. The authors should specify whether patients were censored in follow-up based on the last CT scan obtained, the last plain chest radiograph obtained, or some other criterion. Ideally, follow-up would only be counted based on the last CT (which would be the "gold standard" test here). There is a footnote to Table 4 that provides some information, but it would be helpful to provide person-years of follow-up based on CT only.

3) Fig 1: Some of the numbers do not add up: 

2nd step to 3rd step: starting with 154 cases of surgical lung specimens, and excluding 76 with multiple nodules, leaves 78 cases not 77. 4th step to 5th step: starting with 63 SPNs, and excluding 13 plus 3 plus 7 leaves 40, not 43. 

4) It would be helpful to include data on initial PET/CT findings, if any patients had this test performed. 

5) 3 patients had "unidentified NTM" in the nodules. Did these patients have granulomatous inflammation and/or acid-fast bacilli noted on histopathology? If not, this raises concern that the NTM were lab contaminants.

6) Table 2: PET-CT and SUV are listed in the legend but not in the table. 

7) Table 3 – legend – medium should be median 

8) 4 patients received TB treatment – why? Surely this implies they were symptomatic in some way? Or was TB isolated from other specimens, even though they were asymptomatic? 

9) Table 4 states that patients received no NTM therapy for > 6months. The text implies received no NTM treatment at all. Please clarify - did some receive NTM therapy for <6 months? 

10) Discussion: You state "4 patients received the transient anti-TB medication after surgical resection, which may be a potential confounder to alter the subsequent outcome." Leave out "the transient" - mean TB treatment was 160 days. 

11) Was HIV status tested in any of the patients? If not, this is something to mention in the results as the conclusions may not be generalisable to an HIV-infected population. While some of the patients included were on steroids, most were not on any active immune modulating treatment - so the conclusions in relation to any form of immunosuppression may also be questioned. 

---

## [Author Response · Author response to Decision Letter 0]

26 Aug 2019

Responses to Editor

Q1. We kindly request you to correct the grammatical errors listed below as well as to address the comments of the reviewers.

Ans: We apologize for the errors and have corrected the grammatical errors in the revised manuscript. We have also addressed the reviewers’ comments, item-by-item. Please see the Responses to Reviewers. Thank you.

Responses to Reviewers

1) The method of case ascertainment was not entirely clear. Based on the figure it seems that the authors examined a database containing all positive cultures for NTM, and then looked for patients with resected solitary lung nodules. Additional detail regarding the exact methods used to identify potential cases would be helpful. 

Ans: Thank you for the comment. We have revised the step-by-step description in Figure 1 in the revised manuscript. The descriptions on case selection process in the main text have also been revised (2nd and 3rd paragraphs of the methods section). 

2) As this was a retrospective cohort study, follow-up methods were not standardized for all patients, which may have led to under-ascertainment of recurrent nodules. The authors should specify whether patients were censored in follow-up based on the last CT scan obtained, the last plain chest radiograph obtained, or some other criterion. Ideally, follow-up would only be counted based on the last CT (which would be the "gold standard" test here). There is a footnote to Table 4 that provides some information, but it would be helpful to provide person-years of follow-up based on CT only.

Ans: We thank for the instructive comment. The definition of follow-up duration was added in the methodology (Page 8, Line 1-4). The durations of chest CT follow-up, chest X-ray follow-up and clinical follow-up were also added in the Table 4 of the revised manuscript. 

3) Fig 1: Some of the numbers do not add up: 2nd step to 3rd step: starting with 154 cases of surgical lung specimens, and excluding 76 with multiple nodules, leaves 78 cases not 77. 4th step to 5th step: starting with 63 SPNs, and excluding 13 plus 3 plus 7 leaves 40, not 43. 

Ans: Thank you for the reminder and we apologize for the confused numbers in this figure. We repeatedly counted the 3 cancer patients in the group with pulmonary symptoms. We had corrected the numbers of Figure 1 in the revised manuscript.

4) It would be helpful to include data on initial PET/CT findings, if any patients had this test performed. 

Ans: Thank you for the nice suggestion. Positron emission tomography (PET) was performed in 16 patients, with the maximal standardised uptake value (SUVmax) being 6.4 ± 7.2 (median: 4.2; IQR: 2.0–5.9). We have provided the information in the Results section (Page 12, Line 16-18) and Table 2.

5) 3 patients had "unidentified NTM" in the nodules. Did these patients have granulomatous inflammation and/or acid-fast bacilli noted on histopathology? If not, this raises concern that the NTM were lab contaminants.

Ans: We apologize for the misleading. The histological reports of the 3 resected SPNs all indicated granulomatous inflammation. Tissue mycobacterial culture yielded photochromogens in 2 and nonphotochromogens in the remaining one. However, species identification was not possible under routine laboratory procedures. We have footnoted this information in Table 2 of the revised manuscript. Thank you for the comment.

6) Table 2: PET-CT and SUV are listed in the legend but not in the table. 

Ans: Thank you for the reminder. As suggested by the reviewer in the 4th comment, we have added the information on PET-CT in Table 2 of the revised manuscript.

7) Table 3 – legend – medium should be median 

Ans: We apologize for the typo and have corrected it in the revised manuscript. Thank you. 

8) 4 patients received TB treatment – why? Surely this implies they were symptomatic in some way? Or was TB isolated from other specimens, even though they were asymptomatic? 

Ans: Thank you for the thought-provoking comment. For the 4 patients, the primary care physicians decided to complete a full course of anti-TB treatment based on the histological finding of granulomatous inflammation and positive acid-fast smear of resected SPNs, even knowing that patients were asymptomatic and culture of SPNs yielded NTM. We have footnoted this information in Table 4 of the revised manuscript. 

9) Table 4 states that patients received no NTM therapy for > 6months. The text implies received no NTM treatment at all. Please clarify - did some receive NTM therapy for <6 months? 

Ans: We apologize for the confusing statement. We have revised the descriptions in Table 4 and in the Results section (Page 17, Para 1) as “None of the patients received effective anti-NTM treatment for more than 6 months”. 

10) Discussion: You state "4 patients received the transient anti-TB medication after surgical resection, which may be a potential confounder to alter the subsequent outcome." Leave out "the transient" - mean TB treatment was 160 days. 

Ans: Thank you for the comment. We have deleted the incorrect wording “transient”.

11) Was HIV status tested in any of the patients? If not, this is something to mention in the results as the conclusions may not be generalisable to an HIV-infected population. While some of the patients included were on steroids, most were not on any active immune modulating treatment - so the conclusions in relation to any form of immunosuppression may also be questioned. 

Ans: Thank you for the instructive comments. In this retrospective study, none of the 43 patients received serology test for HIV infection (mentioned in Page 10, Line 5-6). We totally agree with you that the impact of immunosuppression, either disease-associated or iatrogenic, on the risk and clinical course of NTM-SPN remains to be studied. We have added the limitation in the Discussion section (Page 24, Line 12-15).

---

## [Editor Report · Decision Letter 1]

29 Aug 2019

Surgical resection is sufficient for incidentally discovered solitary pulmonary nodule caused by nontuberculous mycobacteria in asymptomatic patients

PONE-D-19-16108R1

Dear Dr. Chong

We are pleased to inform you that your manuscript has been judged scientifically suitable for publication and will be formally accepted for publication once it complies with all outstanding technical requirements.

With kind regards,

Adriano Gianmaria Duse, MD

Academic Editor

PLOS ONE

Additional Editor Comments (optional):

All the recommended changes have been addressed adequately in the revised document.
---

## [Editor Report · Acceptance letter]

4 Sep 2019

PONE-D-19-16108R1 

Surgical resection is sufficient for incidentally discovered solitary pulmonary nodule caused by nontuberculous mycobacteria in asymptomatic patients 

Dear Dr. Chong:

I am pleased to inform you that your manuscript has been deemed suitable for publication in PLOS ONE. Congratulations! Your manuscript is now with our production department. 

With kind regards,

on behalf of

Dr. Adriano Gianmaria Duse 

Academic Editor

PLOS ONE